# Scaling of Average Avalanche Shapes for Acoustic Emission during Jerky Motion of Single Twin Boundary in Single-Crystalline Ni_2_MnGa

**DOI:** 10.3390/ma16052089

**Published:** 2023-03-03

**Authors:** László Z. Tóth, Emil Bronstein, Lajos Daróczi, Doron Shilo, Dezső L. Beke

**Affiliations:** 1Department of Solid State Physics, University of Debrecen, P.O. Box 400, H-4002 Debrecen, Hungary; 2Faculty of Mechanical Engineering, Technion—Israel Institute of Technology, Haifa 3200003, Israel

**Keywords:** twin boundary motion, acoustic and magnetic emission, scaling relations, temporal shapes of avalanches

## Abstract

Temporal average shapes of crackling noise avalanches, U(t) (U is the detected parameter proportional to the interface velocity), have self-similar behavior, and it is expected that by appropriate normalization, they can be scaled together according to a universal scaling function. There are also universal scaling relations between the avalanche parameters (amplitude, A, energy, E, size (area), S, and duration, T), which in the mean field theory (MFT) have the form $E\propto A^{3}$, $S\propto A^{2}$, $S\propto T^{2}$. Recently, it turned out that normalizing the theoretically predicted average U(t) function at a fixed size, $U\left(t\right)=atexp^{-bt^{2}}$ (a and b are non-universal, material-dependent constants) by A and the rising time, R, a universal function can be obtained for acoustic emission (AE) avalanches emitted during interface motions in martensitic transformations, using the relation $R\sim A^{1-\varphi }$ too, where φ is a mechanism-dependent constant. It was shown that φ also appears in the scaling relations $E\sim A^{3-\varphi }$ and $S\sim A^{2-\varphi }$, in accordance with the enigma for AE, that the above exponents are close to 2 and 1, respectively (in the MFT limit, i.e., with φ= 0, they are 3 and 2, respectively). In this paper, we analyze these properties for acoustic emission measurements carried out during the jerky motion of a single twin boundary in a Ni_50_Mn_28.5_Ga_21.5_ single crystal during slow compression. We show that calculating from the above-mentioned relations and normalizing the time axis of the average avalanche shapes with $A^{1-\varphi }$, and the voltage axis with A, the averaged avalanche shapes for the fixed area are well scaled together for different size ranges. These have similar universal shapes as those obtained for the intermittent motion of austenite/martensite interfaces in two different shape memory alloys. The averaged shapes for a fixed duration, although they could be acceptably scaled together, showed a strong positive asymmetry (the avalanches decelerate much slower than they accelerate) and thus did not show a shape reminiscent of an inverted parabola, predicted by the MFT. For comparison, the above scaling exponents were also calculated from simultaneously measured magnetic emission data. It was obtained that the φ values are in accordance with theoretical predictions going beyond the MFT, but the AE results for *φ* are characteristically different from these, supporting that the well-known enigma for AE is related to this deviation.

## 1. Introduction

It is well-known that different parameters of crackling noise avalanches, such as amplitude (A), energy (E), or size (S), are distributed according to a power law, with various exponent values [1,2,3,4,5]; however, it turned out that these exponents are quite robust and do not change considerably for different mechanisms. Thus, the focus is increasingly on the average temporal shapes of avalanches, i.e., the average U(t) functions for a given size (or duration) range of avalanches [1,2,3,4,5]. The average avalanche shapes were studied for many systems producing crackling noise, such as earthquakes [6] or plastic deformation of metals [7,8]. Nevertheless, the most detailed theoretical and experimental works were carried out on magnetic systems [1,9,10,11,12,13,14].

Regarding the theoretical predictions for average avalanche shapes, it was shown [3,4] that at fixed T duration (U(t│T)) can be described by
(1)$$U\left(t|T\right)=T^{\gamma -1}f\left(t\right)\propto T^{\gamma -1}{\left[\frac{t}{T}\left(1-\frac{t}{T}\right)\right]}^{\gamma -1}\left[1-b\left(\frac{t}{T}-\frac{1}{2}\right)\right]$$
where t is the time variable, γ is the exponent of the size-duration relation $S\propto T^{\gamma }$, b is the asymmetry factor, and f(t) is the scaling function characterizing the average temporal avalanche shape. Thus, on the basis of (1), it is expected that the voltage signal divided by $T^{\gamma -1}$ and the time scale divided by T will provide universal scaled functions with some skewness. In addition, it was demonstrated in [3] for Barkhausen noise from the well-known Alessandro–Beatrice–Bertotti–Montorsi (ABBM) model that not only a skewness of the inverted parabola, described by the parameter b [15], is possible but there can be a gradual flattening of the above-scaled function for increasing T (caused by finite values of the demagnetization factor). This flattening was also experimentally demonstrated for plastic deformation of metallic glasses [16], as well as for metal alloys by localized deformation bands (Portevin–Le Chatelier effect) [17]. On the other hand, it was also shown in [18], from acoustic emission (AE) measurements of microcrack formations during compression of different concrete samples, that there can be strong leftward asymmetry in the shape of the $f^{*}=U\left(\frac{t}{T}\right)/T^{\gamma -1}$ functions because the avalanche accelerated faster than it decelerated. Furthermore, it was found that the $f^{*}$ curves did not collapse on a common curve after rescaling. It was also shown in [18] that the slow decaying part of $f^{*}$ can be well fitted by the power function of t, i.e., $f^{*}\sim t^{-p},$ with *p* = 2.05.

For the averaged temporal shape of avalanches at the fixed area it was obtained theoretically—it can be described by [19,20,21,22]:(2)$$U\left(t\right)=ate^{-{\left(\frac{t}{\tau }\right)}^{2}}.$$

Here a and *τ* are non-universal (material-dependent) constants. τ is the characteristic time of the avalanche decay [18] (in [21] $C=\frac{1}{\tau^{2}}$ was used instead of τ). For the experimental verification, whether an appropriate scaling can indeed lead to good scaling together of the averaged temporal shapes at a fixed area, different scaling can be used. Dividing either both axes by $S^{0.5}$ (based on the MFT power relations, i.e., $A\sim S^{0.5}$ as well as T~A) [16,19], or using the theoretical prediction of [20,21], a two-parametrical scaling, with $S_{m}$ and $\tau_{m}$ (having dimensions of size and time) was applied [14]. In [14], introducing the experimental determination of the appropriate $S_{m}$ and $\tau_{m}$ scaling parameters and using Wiener deconvolution, an excellent agreement with experimental data was obtained on Barkhausen-noise measurements on thin amorphous metallic ribbons, where the effect of eddy currents is negligible. On the other hand, the scaled curves obtained from experimental results of jerky deformation of bulk metallic glasses [16] and of Au and Nb nanocrystals [19], in which both axes were scaled by the same parameter $S^{0.5}$, did not display universal behavior. Thus, it is still an open question whether the expected scaling can be achieved by using only one scaling parameter or not. In our recent paper [23], we have shown that the maximum of U in (2), i.e., A, and the rising time belonging to this maximum (R) are interrelated by the parameter a through A~aR. Assuming and experimentally verifying that a is a power function of A, i.e., $a\sim A^{\varphi }$, the relation $R\sim A^{1-\varphi }$ was obtained. Based on this relation, we proposed a new scaling method in which each axis was normalized differently: U is divided by A and t by $A^{1-\varphi }.$ The new scaling method was employed on acoustic emission data captured during martensitic transformation in two different ferromagnetic shape memory alloys, and excellent universal scaled functions were obtained. At the same time, the well-known enigma for the power exponents of the scaling relations [22] was resolved [23] in the sense that in all the above scaling relations, the differences of the exponents from the MFT values were the same: −φ (φ= 0.8 ± 0.1 for both alloys in both scaling relations between S and A as well as E and A).

There are several factors distorting the experimentally determined characteristic parameters of avalanches [22,23], i.e., the experimental verification of the above-described theoretical results for temporal shapes of avalanches can be difficult. The finite value of the threshold, C, at small signals can lead to deviations from the predicted behavior as well as the transfer properties of the detection system can also cause distortions. For instance, the AE signals depend not only on the properties of the source function but also on the macroscopic vibration (ringing) of the sample [22]. In the case of Barkhausen signals, the eddy currents and demagnetization effects [2,14,16] can also cause problems. The authors of Ref. [22] demonstrated that the duration of AE avalanches is distorted by the transfer function of the system, while the amplitude and the rising time remained practically undistorted. This is why we used these parameters in [23], and thus, in the following analysis, we also take the amplitude, A, and $A^{1-\varphi }$ parameters for searching universal temporal shapes of avalanches.

The primary goal of this paper is to apply the above scaling method to new AE measurements that were obtained during the jerky motion of an individual twin boundary in Ni_2_MnGa single crystal, reported in [24]. In [24], simultaneous measurements of stress and magnetic emissions (ME) were carried out, and an approximately linear relationship between the measured ME voltage and the nm-scale volumes exhibiting twinning transformation during microsecond scale abrupt “avalanche” events was found. However, in [24], the statistical analysis of ME avalanches and their temporal shapes were not investigated. In addition, not only the ME, but AE data were also simultaneously detected (but not reported) in the experimental arrangement described in [24], and these measurements and analysis of these data are described here too. It has to be noted that after the publication of our paper [24], in a recent article [25], the authors also performed acoustic emission measurements for the motion of two different types (type I and II) of single twin boundaries in a similar Ni_2_MnGa single crystal. In [25], the authors investigated the correlation between AE and stress evolution, but an analysis of the avalanche temporal shapes was not made.

## 2. Materials and Methods

A single crystalline Ni_50.0_Mn_28.5_Ga_21.5_ sample (produced by AdaptaMat LTD, Helsinki Finland) with dimensions of 20 mm × 3 mm × 2.5 mm was compressed in the fully martensitic state at room temperature. The compression was performed at a rate of 2 μm/s based on a thermal expansion of an aluminum rod, as detailed in [24]. At the beginning of the experiment, one twin boundary of type I was initiated in the middle of the sample. During the experiment, this single twin boundary was propagated due to the compression of the sample. The jerky motion of the interface was detected by a piezoelectric AE sensor (Micro-100s from Physical Acoustics Corp., Mercer County, NJ, USA) placed at one end of the sample. The easy magnetization directions of the two twins, separated by the single twin boundary, were perpendicular. An external magnetic field (0.08 T) was applied at 45° with respect to the axial direction of the sample to magnetically saturate the twins, thus having only one 90° domain wall at the twin boundary. The change of the magnetization was detected via a magnetic sensor coil (50 turns) wrapped around the sample near the moving twin boundary. Other details of the experimental setup can be found in Ref. [24].

The signals of the acoustic and magnetic sensors were amplified, then recorded simultaneously with a National Instruments PCI-6111 data acquisition (National Instruments, Austin, TX, USA) board at 4 MSa/s/channel sampling rate. A bandpass filter (30 kHz–1 MHz) was applied to the AE signals to eliminate the possible DC components, as well as high-frequency white noise. The intermittently occurring offset of the magnetic signal was removed carefully, as it is described in the supporting information of Ref. [24]. The acoustic and magnetic avalanches were detected with the classical threshold-based method with 40 µs hit definition time [23] for the AE. The obtained acoustic avalanches were further filtered using a clustering algorithm [26,27] to eliminate background vibrations.

## 3. Results

### 3.1. Average Avalanche Shape of AE for Fixed Avalanche Size

The average temporal shape was calculated for various size ranges, as it is illustrated in Figure 1. To determine the average avalanche shapes, different avalanche size ranges (bins) were selected to cover the whole size range, excluding the smallest (too low amplitude/threshold ratio) and the largest (too few avalanches to average) avalanches. At each size range, the avalanches were averaged, then the common avalanche parameters (amplitude, size, energy, rising time) were determined for the average function to perform the scaling. For comparison, first, both axes were normalized by $S^{0.5}$, in accordance with the previous convention of Refs. [16,19] (Figure 1a). Secondly, following the suggestions of [23], the vertical axis was normalized by the amplitude, and the horizontal axis was divided by $A_{\;}^{1-\varphi }$ (Figure 1b). For the latter case, the value of φ was determined from the $S\propto A_{\;}^{2-\varphi }$ and $E\propto A_{\;}^{3-\varphi }$ correlations [23], as it is shown in Figure 2, resulting φ= 0.75 ± 0.03 and φ= 0.94 ± 0.02, respectively. It has to be highlighted that the plotted parameters in Figure 2 (amplitude, size, energy, or rising time) are calculated from the (unnormalized) averaged avalanche profiles, not from the individual avalanches, since the parameter φ is derived using Equation (2), which describes the average avalanche profile. (Thus, the number of points in Figure 2a,b are the same as the number of size ranges used in the averaging of the avalanche shapes.) For comparison, Figure 2d shows the usual E versus A plot. It can be seen that this is also a power function with 2.09 ± 0.02 exponent, in quite good agreement with the value indicated in Figure 2b.

Figure 1a shows that the normalization of both axes by only one parameter ($S^{0.5}$) does not provide a good scaling since the normalized curves do not fall on a common curve. On the other hand, as shown in Figure 1b, scaling the voltage by A and the time axis by $A^{1-\varphi }$, provides a good common curve. The relation between R and $A^{1-\varphi }$ can also be directly checked, as it was performed in [23]. It can be seen in Figure 2c that the value of φ (φ= 0.85) obtained from the $S\propto A_{\;}^{2-\varphi }$ and $E\propto A_{\;}^{3-\varphi }$ correlations and the points of the $R\propto A^{1-\varphi }$ correlation are in acceptable agreement with each other. The experimental verification of the latter relation, due to the relatively short values of the rising times, can be difficult, and this is why we proposed in [23] the determination of φ from the $S\propto A_{\;}^{2-\varphi }$ and $E\propto A_{\;}^{3-\varphi }$ correlations.

We can also check whether the exponent 2 in relation (2) gives a good description of the experimental data or not (see, e.g., the trials in [19] for searching an appropriate exponent). Figure 3a shows the exponentially decaying tail for one of the curves shown in Figure 1b, fitted with exponent 2. It can be seen that the fit is very good, i.e., the exponent in Equation (2) is indeed well approximated by this value. For comparison, Figure 3b shows the same type of fit for the same scaled function for martensitic transformation in Ni_49_Fe_18_Ga_27_Co_6_ single crystal [23], and it can be seen that this fit is also good.

We can also compare the exponents of the amplitude or energy probability density functions with exponents given in [25,28,29] for type I twin boundary (*α =* 2.25 [25] and ε = 1.45 [25,28,29] in the $P\left(A\right)\propto A^{-\alpha }$ as well as in the $P\left(E\right)\propto E^{-\varepsilon }$ relations), and those obtained from the probability distributions determined in our measurements. Figure 4 shows these functions, and it can be seen that it gives *α =* 2.05 and *ε =* 1.50, i.e., the agreement is good. Figure 4c shows the so-called maximum likelihood, ML, estimation [30] of the energy exponent, and this gives ε = 1.51 ± 0.1, and a similar estimation for α resulted in α = 2.1 ± 0.1. The results of the ML fits are in good agreement with the values shown in Figure 4a,b.

The exponents of the scaling relation between E and A can be compared with the results of very recent AE experiments carried out for different crackling noise properties [31,32,33,34,35,36]. It can be concluded that these exponents, belonging to ferroelectric switching in BaTiO_3_ (for ferroelectric switching ε = 2.0 [31,32], or for ferroelastic switching ε = 2.0 [32]), to ferroelectric switching in lead zirconate titanate ceramics (ε = 1.97 ± 0.01 [33] to dislocation motion in a high entropy alloy (ε = 2.0 [34]) and in 316 stainless steel (ε = 2.1 [35]), or to porous collapse in granular Mg-Ho alloys (ε = 1.93 ± 0.1 [36]), are very close to those obtained here. This indicates that these values most probably belong to the same similarity class even if the values are definitely smaller than 3, as expected from the MFT theory (see also the Discussion). It has to be noted that in some cases, different exponents were obtained for two different mechanisms in the same material. (See, e.g., the energy exponents for porous collapse or for dislocation motion in [36].) Moreover, the so-called multi-branching was observed [32,34,35] when the exponents were practically the same for the two different mechanisms, but the prefactors were different. These latter fine details are not considered here, although their understanding certainly represents a challenge and calls for further investigations.

### 3.2. Average Temporal Shapes of AE for Fixed Duration

Figure 5 shows the normalized temporal shapes of avalanches for fixed durations using *A* to normalize U, as well as *R* or $A^{1-\varphi }$ to normalize t. For comparison, Figure 6a shows the temporal shape using the usual scaling as dictated by Equation (1), i.e., using the duration time for both axes. It can be seen that the shape of the curves in Figure 5 and Figure 6a is very far from an inverted parabola, but it is reminiscent of the AE results of [18] on microcracking events in different concrete samples. Despite their approximate common scaling, the curves show a strong leftward asymmetry, which can originate from avalanches that accelerate faster than they decay [18]. The slower power law decay is shown in Figure 6b, and a power exponent *p* ≅ 2.0 was obtained. According to the discussions provided in [18], this part most probably cannot be explained by the intrinsic absorption of the acoustic waves (see also Section 4).

### 3.3. Average Temporal Shapes of ME for Fixed Avalanche Size or Duration

In Section 3.1 and Section 3.2, results obtained from the analysis of AE measurements are shown. In this subsection, we apply the same methodology to avalanches measured by the ME sensor. These avalanches were detected in the same experiments as the AE avalanches analyzed above, i.e., using the same experimental conditions and material [24].

Although in our ME results in [24], we used samples with dimensions dictated by conditions to observe the motion of a single twin boundary of type I with magnetically saturated twins, and thus the well-known conditions for avoiding the effects of eddy currents and demagnetization (very thin and long samples) were not met, it is also useful to show the temporal magnetic avalanche shapes. Thus, Figure 7 and Figure 8 show these functions. For magnetic avalanches, the intrinsic absorption (such as for AE) does not appear, and for both scalings, we can use the traditional reduction: for fixed size, the axes were normalized by $S^{0.5}$ (Figure 7a), and for a fixed duration by T (Figure 8). For comparison, Figure 7b shows the normalized function dividing by A, according to the undistorted $S\propto A^{2}$ correlation (see the MFT prediction [2]). It can be seen that the two functions are indeed very similar, as expected. It can be seen that the curves are also satisfactorily scaled together, but the decaying tail is much more extended compared to the theoretical prediction, Equation (2), [14]. It can also be seen from Figure 8 that the average temporal shapes for fixed duration are also scaled well together. The continuous line shows the best fit, which corresponds to γ = 1.69 and a skewness parameter b = 0.60. These values for γ are in line with other data obtained for short-range interaction models [3,4,14], while the skewness parameter is a bit larger than the theoretically predicted values [14,16]. In this case, the value of γ can also be determined from the usual power relation between the area and duration time [1,2,12]. Figure 9 shows this, and the slope of the fitted straight line gives γ = 1.65. Furthermore, Figure 10a,b show the scaling relations between S and A as well as E and A, with scaling exponents 2.23 and 3.25, respectively.

## 4. Discussion

### 4.1. AE Avalanche Shapes for Fixed Area

We have seen that the averaged temporal shapes at different bins of S, can be very well scaled together using A and $A_{m}^{1-\varphi }$(∝R) for normalizing the voltage and the time scales, respectively (Figure 1b). Here, the *φ = 0.85* value was obtained from the exponents of $S\propto A_{m}^{2-\varphi }$ and $E\propto A_{m}^{3-\varphi }$ correlations (and was also consistent with the value suggested by the $R\propto A^{1-\varphi }$ relation: see Figure 2). In contrast to this, the traditional scaling of both axes by $S^{0.5}$ did not provide a common curve (Figure 1a). These results are in good agreement with the results of [23] obtained on interface motions during martensitic transformations of two different ferromagnetic shape memory alloys. The fact that the value of φ=0.85±0.06 obtained here for the intermittent motion of a single twin boundary in the martensitic state and φ=0.8±0.1 provided in [23] for both alloys suggests that this value is the same for both types of intermittent motion. This can also be supported by comparing the shapes of the normalized functions shown in Figure 1b as well as the curves in Figure 13 of [23]: they are indeed very similar. In addition, Figure 3b shows the fit at large values of the reduced time with exponent 2 in Equation (2) for Ni_49_Fe_18_Ga_27_Co_6_ single crystal (this plot was not shown in [23]), and it can be seen that the fit is good, confirming that the tails in both cases have similar behavior indeed.

As can be seen from Figure 4, the exponents of the probability distribution functions of energy and amplitude are in good agreement with the results of [25,29].

### 4.2. AE Avalanche Shape for Fixed Duration

As it is shown in Figure 5, although the curves belonging to different bins of duration time can be scaled together, the shape of the curves is different from the expected inverted parabola with moderate skewness as predicted by the mean-field theory, MFT. There are theoretical analyses beyond MFT [20,25,37] predicting that at large avalanche durations, the parabolic dependence crosses to a linear dependence. It is often argued [12,18] that such a linear dependence can arise from the superposition of sub-avalanches or can be related to retardation effects (eddy currents and/or viscoelasticity [20]) which can also result even in losing the universal behavior. On the basis of the experimental data in hand, in accordance with the conclusions of [18], it is difficult to provide a unique explanation at present. Nevertheless, comparing our results with those of [18], our scaling for the fixed duration with A and $A^{1-\varphi }$, provided good scaling (see Figure 5), while scaling with T, as also performed in [18], is not good (Figure 6a). On the other hand, the tail region has a very similar slope (with about two, as shown in Figure 6b) and obtained in [18].

### 4.3. Avalanche Shape of Magnetic Emission for Fixed Area and Duration

We have seen in Figure 7 that although the curves were satisfactorily scaled together, as compared to the curves shown in Figure 1b, the decaying part did not follow the expected fast decay. This can be attributed to the effects of eddy currents: it was shown theoretically in [20] that the eddy currents (the effects of which, as was mentioned above, could not be excluded here) can indeed cause such distortion. Another possibility, which was also theoretically analyzed in [20], can be the effect of the overlap of sub-avalanches.

Figure 8 illustrates that the temporal shape of avalanches for the fixed duration can be well scaled together, and the value of *γ* obtained from the fit with Equation (1) and provided by the power relation between S and T (Figure 9) are in good agreement with each other.

### 4.4. Comparison of Results Obtained from Acoustic and Magnetic Emission Measurements

Comparing the temporal shapes of avalanches for AE and ME, obtained both for the fixed area and duration—even if one takes into account the possible distortions caused by the detection systems as we treated above—we have to discuss the fact that the normalized curves for both fixed area and duration are dissimilar. This contradiction arises from the expectation that both emission signals originate from the same process: intermittent motion of the single twin boundary. Thus, their statistical behavior should be similar. This is also supported by the special experimental arrangement used in [24]: during the motion of the twin boundary, only signals due to changes of magnetization resulting from the boundary shift appeared in the detection coil of ME (there was no coupling between the twin boundary shift and switching of the “ordinary” magnetic domains). Similarly, the AE signals occur due to spontaneous elastic energy changes during the twinning re-orientation. Now, it was analyzed in detail in [24] that during ME events, there was a one-to-one correspondence between amplitudes of stress drops and amplitudes of their corresponding ME signals. It was also found that magnetic signals showed the same distribution characteristics belonging to boundary shifts both along micro- as well as nanometer (and even subnanometer) scales. However, it was also concluded in [24] that there were numerous avalanches having shorter durations than the detection capability of the ME sensor. These results can explain the features observed for temporal shapes of ME avalanches in Section 3.3: some of the avalanches could be merged into one larger avalanche, i.e., one such avalanche could contain sub-avalanches, as discussed above. This, together with the possible effects of eddy currents, can explain both the slow decay in Figure 7 and the relatively large value of the asymmetry parameter b obtained from the fit of the curve shown in Figure 8.

For the results obtained from AE measurements, it is worth mentioning that in [25,28], no obvious correlation between the stress drops and the AE signals could be observed: sometimes, the stress drops were not accompanied by AE with relevant magnitude. Furthermore, it was concluded in [29] that the AE signals emitted during stress drops as well as AE signals generated by local microscopic events between stress drops, showed different distribution behavior, and only in the latter case was the AE attributed to a real (but not global) shift of the twin boundary. The above different behavior was also explained by the restricted bandwidth of the AE sensor: for large stress drops, only a small fraction of the radiated elastic energy was detected as the energy of the AE signal. Whether these observations alone can be enough to explain the main deviation of the temporal avalanche shapes at the fixed area or duration is still an open question. In fact, the shape belonging to a fixed area reflects a behavior predicted by Equation (2), while the shape at fixed duration shows a long, slow-decaying tail. Although in [18] some explanation for such behavior was offered, this problem still calls for further investigation. Perhaps detailed investigations of correlations between the waiting times within AE and ME signals as well as the correlations between the simultaneous AE and ME signals, can provide useful information to this problem. Such an analysis is in progress and will be published soon.

### 4.5. On the Compatibility of Scaling Exponent Derived from Expressions for the Average Avalanche Shapes at Fixed Duration and Area

There is one interesting feature of the scaling exponents obtained from simultaneous AE and ME measurement, namely that they show a definite difference from the MFT values for AE, while they differ much more moderately for ME. We can illustrate that the above difference is another manifestation of the well-known enigma of AE, as follows.

These exponents should be the same, starting from expressions valid for a fixed duration or area. Now it was shown in [23] that starting from Equation (2)
(3)A∝aR,
and assuming that a has a power dependence on A ($a\propto A^{\varphi }$)
(4)$$R\propto A^{1-\varphi }$$

As a consequence of (4), using the definitions of E and S [23] and the universal form of the scaled $\frac{U\left(\frac{t}{R}\right)\;}{A}$ function, it was obtained that the power exponents should be decreased by the same value of φ.

For a fixed duration time, we can start from a bit modified version of (1) (see Equation (S3) in the supplement of [4])
(5)$$U\left(\frac{t}{T}\right)\propto T^{\gamma -1}{\left[\frac{t}{T}{\left(1-\frac{t}{T}\right)}^{B}\right]}^{\gamma -1}$$
where the new skewness parameter, B, gives back the symmetric case for B=1, and the asymmetry is negative for B1 (avalanches skew toward to the end). (5) has it maximum at $R=\frac{T}{1+B}$ and thus for the peak amplitude (belonging to $\frac{T}{R}=\frac{1}{1+B}$) we have
(6)$$A\propto R^{\gamma -1}{\left(1-\frac{1}{1+B}\right)}^{B\left(\gamma -1\right)}$$
i.e., for fixed values of B and γ−1, the relation between A and R is
(7)$$A\propto R^{\gamma -1}.$$

Since it is expected that γ is the same for a given mechanism, and it was also guessed in [23] that in (4) φ has the same property, the compatibility condition means that
(8)$$\varphi =\frac{\gamma -2}{\gamma -1}$$

According to theoretical predictions beyond MFT, γ can change between 1.57 and 2.00 [4,21]. Thus, the MFT (with γ=2) gives φ=0, and φ should be negative beyond MFT. It is indeed the case for our data from ME: from the scaling exponents obtained above φ≅−0.25±0.03, and it is also in acceptable agreement with γ=1.69. On the other hand, the value of φ=0.8 obtained from AE data from almost the same source is characteristically different. This means that although in [23] and here we got good scaled together functions for averaged temporal shapes of AE avalanches with the use of the above φ, the enigma still exists in that sense that (8) is not fulfilled for AE data.

## 5. Conclusions

Analyzing simultaneous acoustic and magnetic emission signals measured during the jerky motion of a single twin boundary in a Ni_50_Mn_28.5_Ga_21.5_ single crystal during slow compression, the temporal avalanche shapes, U(t), were constructed for fixed area and duration. Our conclusions are as follows:

(i)Using scaling proposed in [23] (i.e., normalizing the time axis of the average avalanche shapes with $A^{1-\varphi }$, and the voltage axis with A) good common universal functions were obtained for both fixed area and fixed duration. For the fixed area, the obtained functions followed well the theoretically predicted shape (Equation (2)), and the φ value determined from scaling relations between the rising time versus peak amplitude, A, area versus A as well as energy versus A, were the same (φ=0.85±0.06) and agreed well with values found from AE measurements on martensitic transformations in two different ferromagnetic shape memory alloys. For a fixed duration time, although the functions could also be scaled together, the shape was characteristically different from the shape predicted by Equation (1). This very asymmetric behavior, similar to the results of [18], can be described by a slow power-law decay with an exponent of about two.(ii)Investigating the similar scaling behavior for magnetic emission data, it was obtained that the φ values were in accordance with theoretical predictions going beyond the MFT (φ=−0.25). Similarly, the temporal shapes for fixed area and duration were in accordance with expressions (1) and (2), with a small deviation most probably due to eddy currents distortions. On the other hand, the AE and ME results for φ were characteristically different from each other. This supports that the well-known enigma for AE (i.e., the power exponents between E and A as well as between S and A are given by 3−φ as well as 2−φ) is related to this deviation, and it calls for further investigation.

## Figures and Tables

**Figure 1 materials-16-02089-f001:**
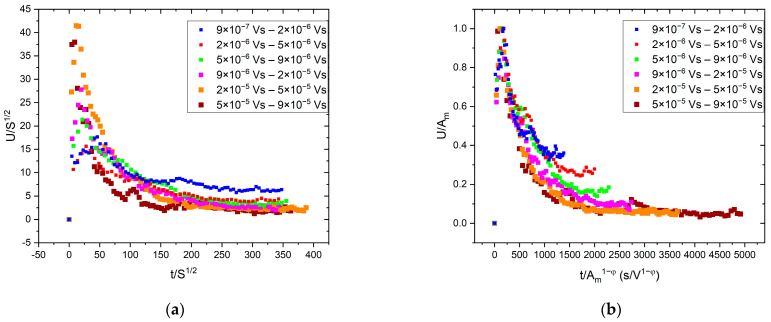
Average temporal shapes of AE avalanches for various fixed avalanche size ranges. (**a**): U/S^1/2^ vs. t/S^1/2^; (**b**): U/A vs. t/A^1−φ^.

**Figure 2 materials-16-02089-f002:**
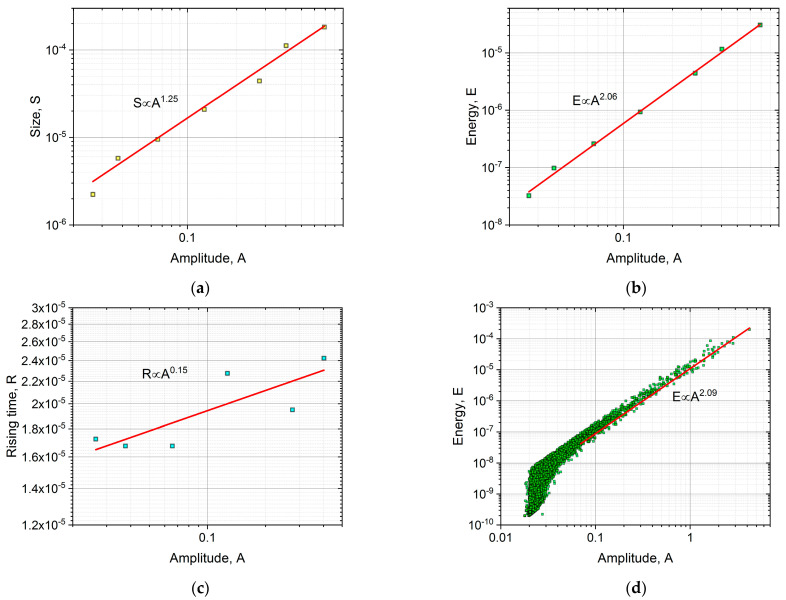
Relations between the parameters of the averaged avalanche profiles, (**a**): $S\propto A_{\;}^{2-\varphi }$, (**b**): $E\propto A_{\;}^{3-\varphi }$, and (**c**): $R\propto A_{\;}^{1-\varphi }$. The φ values from the fittings of (**a**,**b**) are 0.75 ± 0.03 and 0.94 ± 0.02, respectively. The final value, φ= 0.85, was determined as the average value of these two fittings, since the rising time has high uncertainty. The red line in (**c**) indicates the expected slope, using φ = 0.85. For comparison, (**d**) shows the relation between the energy and amplitude of the individual avalanches, resulting φ= 0.91, which is close to the value determined from (**b**).

**Figure 3 materials-16-02089-f003:**
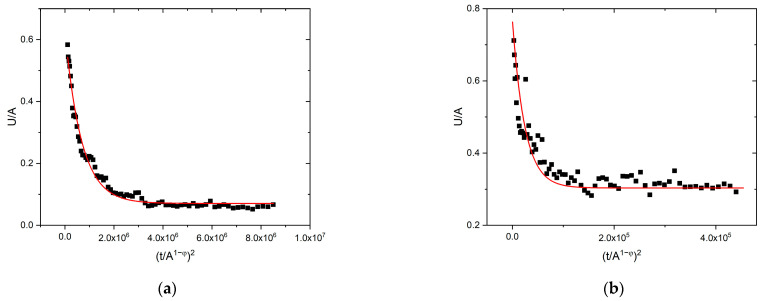
Fitting of the exponentially decaying tail of (**a**): curve 4 of Figure 1b (9 × 10^−6^ Vs–2 × 10^−5^ Vs) and (**b**): curve 5 of Figure 13 in [23] (7.5 × 10^−6^ Vs–8.5 × 10^−5^ Vs), according to Equation (2).

**Figure 4 materials-16-02089-f004:**
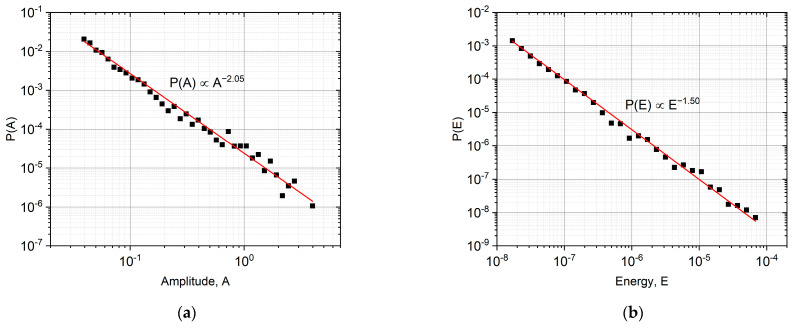
Probability density functions for the avalanche amplitude (**a**) and energy (**b**). (**c**): Maximum likelihood estimation for the energy power exponents.

**Figure 5 materials-16-02089-f005:**
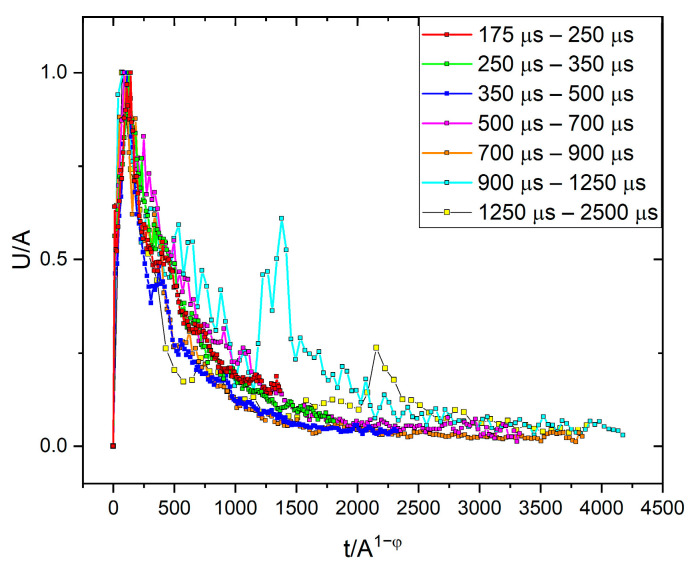
Average temporal shapes of AE avalanches for various fixed duration ranges: U/A vs. $t/A^{1-\varphi }$.

**Figure 6 materials-16-02089-f006:**
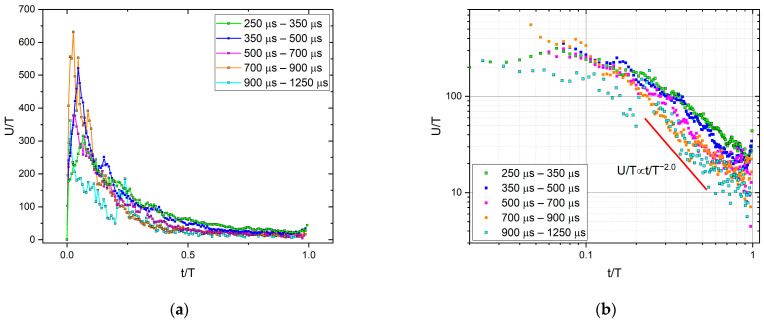
(**a**): Average temporal shapes of AE avalanches for various fixed duration ranges. U/T vs. t/T. (**b**): Fitting of the decaying tail of curves of (**a**).

**Figure 7 materials-16-02089-f007:**
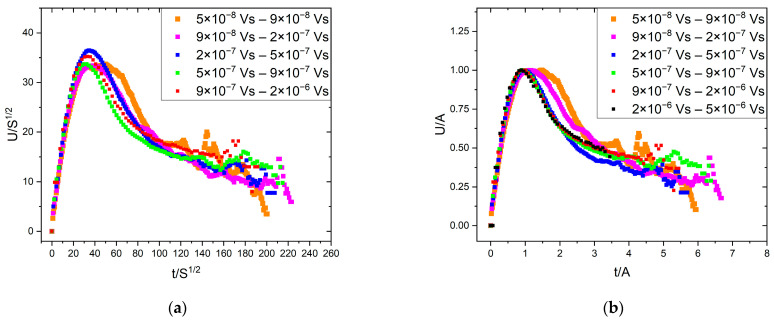
Average temporal shapes of ME avalanches for various fixed avalanche size ranges. (**a**): U/S^1/2^ vs. t/S^1/2^. (**b**): U/A vs. t/A.

**Figure 8 materials-16-02089-f008:**
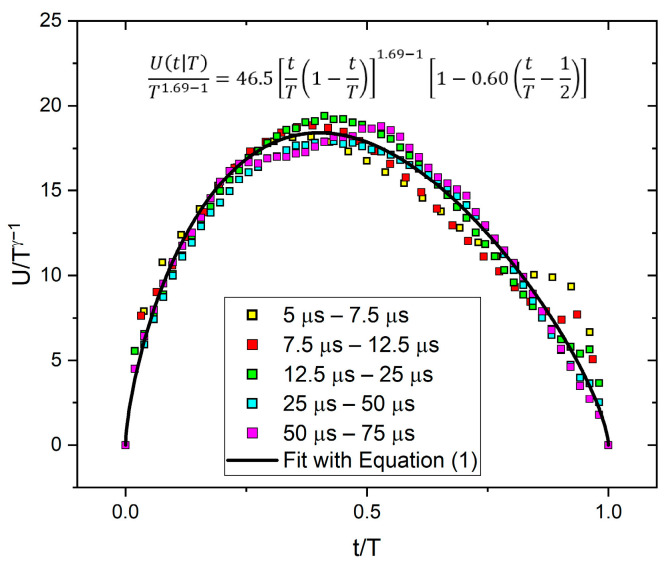
Average temporal shapes of ME avalanche for various fixed duration ranges. The average of the curves was fitted according to Equation (1), resulting γ = 1.69 and b = 0.60.

**Figure 9 materials-16-02089-f009:**
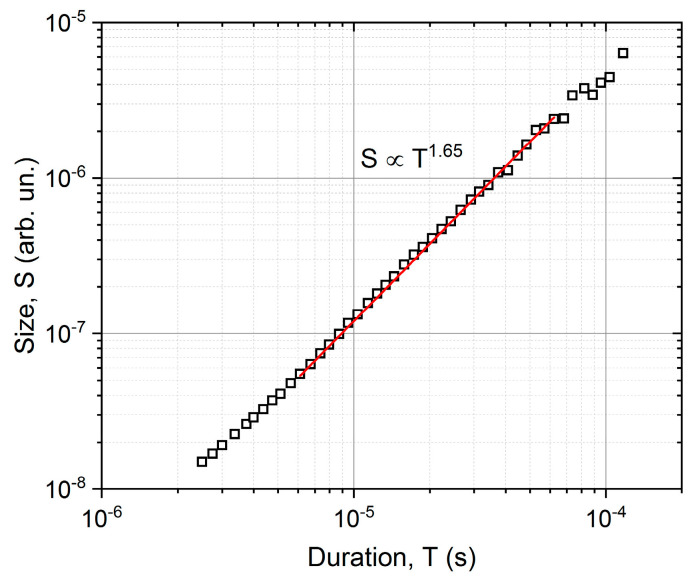
S versus $T_{\;}^{\gamma }$ function, fitted in the range of the average avalanche shape functions, resulting γ = 1.65, which is in very good agreement with the γ value obtained from the fit of the average avalanche shapes.

**Figure 10 materials-16-02089-f010:**
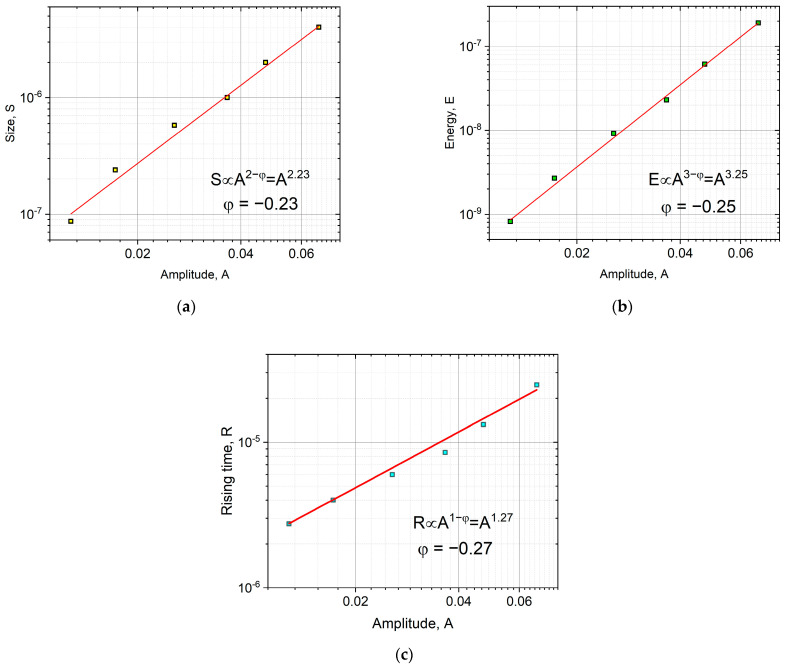
Scaling functions between the (**a**): averaged avalanche area, S, and amplitude, A, (**b**): averaged energy, E, and amplitude; as well as (**c**): averaged rising time, R, and amplitude. The φ values from the fittings of (**a**–**c**) are −0.23 ± 0.03, −0.25 ± 0.02, and −0.27 ± 0.02, respectively.

## Data Availability

The data that support the findings of this study are available from the corresponding author upon reasonable request.

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
