# Peer review of "Scaling of Average Avalanche Shapes for Acoustic Emission during Jerky Motion of Single Twin Boundary in Single-Crystalline Ni2MnGa"

_materials, 2023, doi:10.3390/ma16052089_

Round 1

Reviewer 1 Report

I like the study and advocate publication. The results are important and the field is in full development. There are some points that I would like to bring to the attention of the authors, mainly to make the paper more relevant to a larger community and perhaps calibrate the findings. 

The paper is hard to understand for a new reader without reading first the previous papers [23], [24] [25]. This is OK if a major new conclusion is presented but, ultimately, this depends on the quality of the data. Taken as an AE study, the data are very good ( and certainly good enough to be published as such). The paper endeavours to show something much more far-reaching, however, namely that a scaling approach using a new scaling parameter is useful (with which I agree).  This is then a matter of very precise data to argue this case. As the authors say themselves, this ultimate goal was not reached, although the paper represents a good step in this direction. Actually, I was left with the impression that the paper could almost mean the opposite, namely that the proposed scaling is not that clear at all. 

Experimentally, the best data set stems from the energy~amplitude scaling Fig.2b. The other fits are really not good enough to decide such subtleties like the actual value of phi. Nevertheless, the general approach is correct  but the data spread is too big. The key parameter in Fig.2b is the exponent X= 3-phi. The authors insist on high precision values of X which is indeed needed to make more fundamental claims. X=2 occurs for very narrow avalanche functions and increases to 3 with broader avalanche functions. This brings it down to the question: what is the exact value of X. In Fig. 2b one may argue that X=2 is correct within the error margins. The claimed value of 2.06 may be OK but that is not conclusively shown and the data do scatter. A more careful error analysis (e.g. ML based?) is clearly desirable. It is then argued that the deviation from 2 is bigger in other measurements but there are not stringent error estimates given. Some of the fits are simply not convincing, but probably the best that can be done. 

I am fully aware that the required data are very hard to measure and I think that authors did a good job - and only some better estimates + discussion is needed to clear this point.

However, there are other data sets in literature, some of which are much better quality. I would think that citing these data would give the paper much more weight. I just list some of them below, I know that others are known to the authors. The best data set is probably 

E. K. H. Salje, D. Xue, X. Ding, K. A. Dahmen, and J. F. Scott
Phys. Rev. Materials 3, 014415 – Published 28 January 2019

with X=2.0 The error may be +-0.01. In PHYSICAL REVIEW MATERIALS 6, 124413 (2022), the fits  with X=2 which were excellent for high energies but may deviate slightly at low energies. In this case, the non-avalanche noise is relevant and can change the exponents if the data are measured in a restricted energy interval (as done also in this paper). The same bending is seen in some data in Journal of Applied Physics 132, 080901 (2022); https://doi.org/10.1063/5.0098813 so that a fit was not attempted.

A value of 1.97+- 0.01 was reported inJ. Appl. Phys. 132, 224102 (2022); https://doi.org/10.1063/5.0126308  2022. As this is smaller than 2 there may be a different mechanism at play.

Values of X= 1.6, 1.7,1.9 and several of 2 are in Scientific Reports | (2019) 9:1330 | https://doi.org/10.1038/s41598-018-37604-5. Here the small values are all for small energies while all high energy data show x=2. Interestingly, all data for energies > 10 aJ show X=2 with high precision, the smaller values are again below 2.

Values of 2.1 and 2 are shown in Appl. Phys. Lett. 117, 262901 (2020); https://doi.org/10.1063/5.0030508  2020.

I am sure that there must be more data in particular for 'multi branching' where these exponents are derived as a matter of course because the entire technique relies on the non-universal prefactor (and not on the exponent itself) while the exponents need to be fitted to obtain the refactor. I

t appears that all data are close to 2 but deviate for smaller energies. If anything , this makes the analysis difficult as reported in this paper.

I recommend to the authors to add a brief discussion and add the references to previous exponents. If they could identify the reason why the low energy data are 'damaged' it would make the paper really very important.

Author Response

Dear Reviewer,

First of all thank you very much for your valuable comments, which were taken into account during the revision of our paper.

Our answers to your comments on the quality of the data:

We presented the scaling relations between the area and amplitude as well as energy and amplitude using the averaged (within a certain bin of the area) values instead of the usual plots where all measured points are included. For comparison we inserted this plot for the E versus A relation in Fig.2d and as it can be seen the plot and the error bar of the fitted exponent is very similar to those you cited from other recent papers. We also gave the error bars for the other calculated values, as well as presented a ML analysis for the P(E) as well as P(A) distributions (see the new figure 4c).  

In addition, accepting that inclusion recent papers on AE measurements will increase the weight of the paper, we inserted a new paragraph at the end of Chapter 3.1 and cited the articles listed in your comments. We mentioned here that in our opinion the multi-branching is rather related to the presence of two different mechanisms and as such (since we are quite sure that we had only one well defined mechanism in our case) was not considered further in our discussion. We agree that identifying the reason of it is an important challenge for further investigations.

We hope that, after corrections made, the paper can be accepted for publication.

Reviewer 2 Report

The main goal of this work is to apply the above scaling method on new AE measurements that were obtained during jerky motion of an individual twin boundary in Ni2MnGa single crystal. I think the authors sucessed to illustarated that point. Thus, I recommend to accept the current manuscript after doing minor revsion.

1- English should checked by native speaker as there are some parts in the introduction and the disscousion are not written very well.

2- the authors should add the doi of the references cited in the current manuscript.

3- Could the authors add more recently references related to the current work.

4- If possible, could the authors adding a supporting information parts to declare thier calculation and the method they used in the current work.

Author Response

Dear Reviewer,

First of all thank you very much for your valuable comments, which were taken into account during the revision of our paper.

Our answers to your comments are as follows:

  • We made a careful check of the English
  • Doi references were added
  • We included 6 new references of very recent papers on AE investigations ( see new references 32-37
  • Since the experimental arrangement and the details of the magnetic signal detection was described in our previous paper (as cited in the text) we described here only the way of the detection of the AE signals.

We hope, that after corrections made, the paper can be accepted for publication.